# LGR2: Language Guided Reward Relabeling for Accelerating Hierarchical Reinforcement Learning

## Abstract

Large language models (LLMs) have shown remarkable abilities in logical reasoning, in-context learning, and code generation. However, translating natural language instructions into effective robotic control policies remains a significant challenge, especially for tasks requiring long-horizon planning and operating under sparse reward conditions. Hierarchical Reinforcement Learning (HRL) provides a natural framework to address this challenge in robotics; however, it typically suffers from non-stationarity caused by the changing behavior of the lower-level policy during training, destabilizing higher-level policy learning. We introduce LGR2, a novel HRL framework that leverages LLMs to generate language-guided reward functions for the higher-level policy. By decoupling high-level reward generation from low-level policy changes, LGR2 fundamentally mitigates the non-stationarity problem in off-policy HRL, enabling stable and efficient learning. To further enhance sample efficiency in sparse environments, we integrate goal-conditioned hindsight experience relabeling. Extensive experiments across simulated and real-world robotic navigation and manipulation tasks demonstrate LGR2 outperforms both hierarchical and non-hierarchical baselines, achieving over 55% success rates on challenging tasks and robust transfer to real robots, without additional fine-tuning.

## 1 Introduction

Robotic systems capable of understanding and executing natural language instructions hold great promise for enabling intuitive human-robot interaction and flexible automation. Recent advances in deep reinforcement learning (RL) have demonstrated remarkable success in learning complex robotic behaviors from raw sensory inputs (Levine et al., 2015; Kalashnikov et al., 2018; Rajeswaran et al., 2017). However, these methods often struggle with long-horizon tasks that require extensive planning and precise multi-step coordination, particularly under sparse reward signals where meaningful feedback is rare and delayed.

Hierarchical reinforcement learning (HRL) provides a principled framework to tackle these challenges by decomposing tasks into temporally extended subgoals and learning nested policies (Sutton et al., 1999; Dayan & Hinton, 1993; Vezhnevets et al., 2017). This temporal abstraction improves exploration and credit assignment, essential for solving complex robotic control problems. Yet, conventional HRL algorithms face a significant hurdle: *non-stationarity*. As the lower-level policy evolves, the higher-level policy encounters shifting dynamics and reward distributions, destabilizing training and impeding convergence (Levy et al., 2018; Nachum et al., 2018).

Recent advances in large-scale language models (LLMs) present an exciting opportunity to address these challenges. LLMs excel at processing and generating meaningful representations from natural language (Brown et al., 2020; Wei et al., 2022; Kojima et al., 2022), and have been increasingly applied to robotics for tasks like instruction grounding and code generation (Liang et al., 2023b; Ahn et al., 2022; Huang et al., 2022). Approaches that translate natural language commands into reward functions (L2R) have shown promise (Yu et al., 2023; Sharma et al., 2022; Kwon et al., 2023). However, prior L2R methods often target monolithic policies and lack the hierarchical structure needed to efficiently solve long-horizon, sparse reward problems.

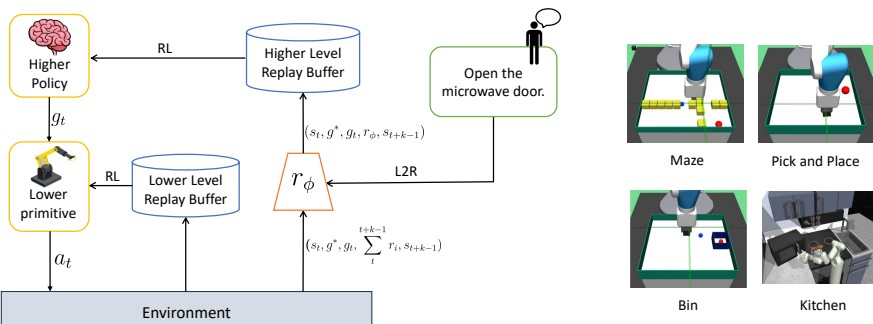

Figure 1: **LGR2 overview (left):** The higher-level policy predicts subgoals $g_t$ for the lower-level policy, which executes primitive actions $a_t$ on the environment. The lower-level replay buffer is populated by environment interactions, and lower-level policy is optimized by RL. L2R is used to translate human instructions to reward function parameters, which subsequently relabel the higher-level replay buffer transitions. Finally, RL is used to optimize the higher-level policy.
**Environments (right):** maze navigation, pick and place, bin, and franka kitchen.

In this work, we propose **LGR2**, a novel HRL framework that harnesses LLMs to generate *language-guided reward functions* for the higher-level policy. Unlike vanilla HRL approaches, LGR2 employs a novel two-stage pipeline: first using LLMs as motion descriptors to convert natural language instructions into structured, canonical task representations, and then employing reward coders to generate symbolic, executable reward parameters that remain invariant to policy changes. By translating natural language instructions into symbolic and semantically rich reward parameters, LGR2 effectively decouples the high-level reward evaluation from the evolving lower-level behaviors, substantially reducing non-stationarity while maintaining semantic alignment with human intentions.

Our approach fundamentally differs from prior language-to-reward (L2R) Yu et al. (2023) methods by introducing hierarchical temporal abstraction, enabling efficient decomposition of complex, long-horizon tasks into manageable subgoals. Further, in order to address reward sparsity common in challenging environments, we incorporate goal-conditioned hindsight experience replay (HER) Andrychowicz et al. (2017) to densify and enrich the high-level reward signal. This synergistic integration of hierarchical decomposition and reward densification creates a robust framework to stabilize HRL training.

Our key contributions are as follows:

1. We introduce LGR2, an end-to-end framework that uses LLM-based reward generation to guide hierarchical policies to solve complex robotic tasks using natural language instructions, thus mitigating the non-stationarity issue in HRL.

2. We demonstrate that the language-guided reward relabeling scheme effectively stabilizes the higher-level policy training by achieving a stationary reward signal, thereby addressing key causes of HRL instability.

3. We incorporate HER to combat sparse rewards, significantly improving sample efficiency and generalization.

4. Through extensive experiments on challenging simulated and real-world robotic navigation and manipulation tasks, we establish that LGR2 achieves more than 55% higher success rates over strong hierarchical and flat baselines, and achieves robust zero-shot transfer to physical robots without additional fine-tuning.

## 2 RELATED WORK

**Hierarchical Reinforcement Learning.** HRL has been extensively studied as a promising approach to address the challenges of long-horizon tasks by decomposing complex behaviors into multiple levels of temporal abstraction (Barto & Mahadevan, 2003; Sutton et al., 1999; Parr & Russell, 1998; Dietterich, 1999). However, a fundamental challenge in off-policy HRL is *non-stationarity* that destabilizes HRL training (Levy et al., 2018; Nachum et al., 2018). Existing solutions attempt to alleviate this problem by simulating expert lower-level behaviors (Levy et al., 2018), relabeling replay

buffers (Nachum et al., 2018), or leveraging privileged information such as demonstrations (Gupta et al., 2019; Singh & Namboodiri, 2023b;a) or preferences (Singh et al., 2024b;a). In contrast, our work proposes LGR2, which uniquely harnesses LLMs to generate stable, language-guided reward parameters for the higher-level policy. By decoupling high-level reward generation from evolving low-level policies, LGR2 effectively mitigates non-stationarity in off-policy HRL, thereby enabling more robust and efficient learning.

**Language to Actions.** Early work in language-conditioned robotics mapped structured natural language commands to controllers using temporal logic (Kress-Gazit & Pappas, 2008) or motion primitive parsing (Matuszek et al., 2012). More recent end-to-end models translate natural language instructions into robot actions, especially for navigation (Ku et al., 2020), but often assume low-dimensional discrete action spaces (e.g., moving between graph nodes) (Ku et al., 2020; Kamath et al., 2023) and require extensive training data.

Latent language embeddings trained with behavioral cloning (Mees et al., 2023; Jang et al., 2022; Lynch et al., 2022), offline RL (Ebert et al., 2021), goal-conditioned RL (Fu et al., 2019), or shared autonomy (Karamcheti et al., 2021) have been employed to condition policies on natural language commands. Despite their promise, these end-to-end models require vast data and struggle with long-term planning. Recently, Yu et al. (2023) proposed a reward-based method where an optimal controller generates low-level actions, reducing data needs. Our work extends this by incorporating temporal abstraction through HRL, improving training efficiency and handling complex, long-horizon tasks effectively.

**Language to Code.** Large language models such as LLaMA (Touvron et al., 2023), GPT-4 (OpenAI et al., 2024), and Gemini (Team et al., 2024) have revolutionized code generation capabilities, enabling applications ranging from competitive programming (Li et al., 2022) and drawing (Tian et al., 2020) to policy synthesis for 2D tasks and complex instructions (Trivedi et al., 2022; Liang et al., 2023a). We leverage these models' coding and reasoning abilities to generate language-guided, higher-level reward functions that facilitate learning in long-horizon robotic control tasks.

**Language to Rewards.** Translating natural language instructions into reward functions has been explored in recent work (Sharma et al., 2022; Goyal et al., 2019; Nair et al., 2022; Bahdanau et al., 2018; Hu & Sadigh, 2023; Kwon et al., 2023; Lin et al., 2022a). Many such methods rely on training domain-specific reward models that map instructions to reward signals or constraints (Sharma et al., 2022; Goyal et al., 2019; Nair et al., 2022). Although effective on certain tasks (e.g., object pushing, drawer opening), these approaches necessitate large quantities of annotated language-reward data.

Recent advances explore using LLMs to infer user intents and assign rewards in interactive or game-based settings (Kwon et al., 2023; Hu & Sadigh, 2023), yet applying LLMs for real-time reward assignment during RL remains limited due to high query costs. AutoRL (Chiang et al., 2018) introduced automated parameterization of reward functions but lacked a natural language interface. Distinctively, our work leverages language to generate reward parameters that directly relabel higher-level replay buffers in HRL, enabling efficient training without relying on massive labeled datasets or expensive online LLM querying.

## 3  PROBLEM FORMULATION

**Hierarchical Setup.**

We model the robotic control task as a goal-conditioned Markov Decision Process (MDP) defined by the tuple $\mathcal{M} = (\mathcal{S}, \mathcal{A}, p, r, \gamma, \mathcal{G})$, where $\mathcal{S}$ denotes the state space, $\mathcal{A}$ the action space, $p(s' \mid s, a)$ the transition dynamics, $r : \mathcal{S} \times \mathcal{A} \to \mathbb{R}$ the reward function, $\mathcal{G}$ the goal space, and $\gamma \in (0, 1)$ the discount factor. The policy $\pi$ maps states and goals to distributions over actions. In the goal-conditioned setup, the policy is conditioned on both the current state $s_t$ and a desired goal $g_t$, such that $a_t \sim \pi(\cdot \mid s_t, g_t)$.

We adopt a two-level hierarchical reinforcement learning (HRL) framework, where a higher-level policy $\pi_H : \mathcal{S} \to \Delta(\mathcal{G})$ selects subgoals $g_t \in \mathcal{G} \subseteq \mathcal{S}$ every $k$ environment steps, and a lower-level policy $\pi_L : \mathcal{S} \times \mathcal{G} \to \Delta(\mathcal{A})$ conditions on the current state and subgoal $g_t$ to execute $k$ primitive actions aimed at achieving $g_t$.

At $k$-step intervals, the higher-level policy stores experience tuples $\Sigma_t = (s_t, g^\star, g_t, r_t^H, s_{t+k})$, where $s_t$ is the current state, $g^\star \in \mathcal{G}$ the final user-specified goal, $g_t$ the subgoal chosen by the higher-level

policy, $r_t^H = \sum_{i=0}^{k-1} r(s_{t+i}, a_{t+i})$ the cumulative environment reward over the $k$ lower-level steps, and $s_{t+k}$ the next state after executing these actions.

Similarly, the lower-level policy stores transitions in its replay buffer in the form: $(s_t, g_t, a_t, r_t^L, s_{t+1})$, where $r_t^L$ is typically a reward encouraging progress towards the subgoal $g_t$ (e.g., $r_t^L = -\mathbf{1}_{\{\|s_t - g_t\| > \varepsilon\}}$). The hierarchical structure enables temporal abstraction, where the higher-level policy focuses on strategic subgoal selection, and the lower-level policy handles execution details over shorter time horizons.

**Language to Rewards.** Recent work has explored translating natural language instructions into reward functions to guide RL (Sharma et al., 2022; Lin et al., 2022b; Kwon et al., 2023). These approaches typically rely on training domain-specific reward models that interpret language commands and produce corresponding reward signals. Notably, Yu et al. (2023) introduce a modular *reward translator* framework comprising two key components: a *motion descriptor*, which converts natural language instructions into structured action descriptions, and a *reward coder*, which maps these descriptions into concrete reward function parameters. This pipeline enables efficient interfacing between high-level language commands and the low-level robotic control primitives, facilitating end-to-end task specification without requiring task-specific reward engineering.

**Challenges.** Despite significant progress in using language to specify robotic objectives, several core limitations persist in the context of complex, long-horizon tasks:

**Limitations of Language-to-Reward (L2R) Approaches.** Existing L2R methods face three principal obstacles. First, learning a single-level policy for long-horizon tasks is fundamentally difficult due to long-term credit assignment: sparse or delayed rewards make it hard for the agent to attribute success or failure to specific actions taken many steps prior. Second, reward functions automatically synthesized from natural language instructions are often highly sparse, providing insufficient learning signal for the lower-level policy to make meaningful progress, especially in tasks requiring intricate sequences of behaviors. Third, many prior approaches rely on either pre-defined or manually-engineered libraries of control primitives to bridge the gap between abstract language instructions and executable actions. This not only demands significant expert knowledge and engineering effort, but also reduces scalability and adaptability to new tasks or environments.

**Non-Stationarity in Off-Policy HRL.** A further critical challenge arises in hierarchical reinforcement learning, especially in off-policy settings: the problem of *non-stationarity*. As the lower-level policy $\pi_L^{(m)}$ evolves over the course of training (with $m$ indexing training iteration), the conditional distribution over next states $s_{t+k}$ after executing a fixed high-level subgoal $g_t$ changes:

$$p_{g_t}^{(m)}(s_{t+k} \mid s_t) = \Pr\big(s_{t+k} \mid s_t, g_t, \pi_L^{(m)}\big).$$

Accordingly, the cumulative high-level reward over the $k$-step interval,

$$r_t^{H(m)} = r^H(s_t, g^\star, g_t; \pi_L^{(m)}),$$

becomes non-stationary, as it depends on the continually adapting lower-level behavior. This drift invalidates older transitions stored in the higher-level replay buffer, undermining the stability and effectiveness of off-policy learning for the higher-level policy $\pi_H$ and substantially impeding convergence. These intertwined challenges: $(i)$ lack of temporal abstraction and compositionality in L2R methods, $(ii)$ reward sparsity, $(iii)$ the inflexibility of manual skill libraries, and $(iv)$ non-stationarity in hierarchical RL, motivate the need for new frameworks that combine the semantic richness of language guidance with scalable, stable hierarchical learning for complex robotic control.

## 4 METHODOLOGY

We address the non-stationarity problem in HRL by introducing *language-guided reward relabeling*, which produces a stationary and semantically meaningful high-level reward function independent of the evolving lower-level policy.

### 4.1 REWARD PARAMETER GENERATION

Given a natural language instruction $\mathcal{L}$ provided by the user, a Large Language Model (LLM) is employed to generate a structured representation of the task and subsequently translate it into reward

function parameters. Inspired from L2R framework Yu et al. (2023), this is implemented via a two-stage pipeline using a motion descriptor and a reward coder module. This established pipeline operates as follows:

**1. Motion Descriptor Module:** Given a user's natural language instruction $\mathcal{L}$, this module uses large language models (LLMs) to generate a structured task description $d = \mathcal{M}_{\text{desc}}(\mathcal{L})$ in templated natural language. This step transforms ambiguous instructions into clear, canonical representations that facilitate downstream processing (Yu et al., 2023).

**2. Reward Coder Module:** Conditioned on the structured motion descriptor $d$, the reward coder uses the LLM's code generation capabilities to output parameters $\phi = \mathcal{M}_{\text{code}}(d)$ for a symbolic, goal-conditioned reward function. The coder produces explicit, executable code (or interpretable parameter sets) that define the reward evaluation logic for the specified skill.

Formally, the language-conditioned reward function is defined as $r_\phi : \mathcal{S} \times \mathcal{G} \times \mathcal{G} \to \mathbb{R}$, where each triple $(s, g^\star, g)$ consists of the current environment state $s \in \mathcal{S}$, the final user-specified goal $g^\star \in \mathcal{G}$, and a candidate subgoal $g \in \mathcal{G}$ proposed by the higher-level policy. The higher level reward function $r^H$ evaluates

$$r^H = r_\phi(s, g^\star, g) = f(s, g^\star, g; \phi),$$

where $f$ is a parameterized function with fixed parameters $\phi$ generated by the LLM-based reward coder. Note that the reward function $r_\phi$ depends exclusively on the current state and goal variables, and is independent of the low-level policy. This decoupling effectively mitigates non-stationarity in HRL by providing a stable, consistent reward signal. Consequently, reward assignment remains stationary, semantically meaningful, and highly modular, which are essential for reliable and efficient hierarchical learning in complex, long-horizon robotic control tasks.

## 4.2 RELABELING OF HIGH-LEVEL REWARDS

Each stored transition in the higher-level replay buffer,

$$\Sigma_t = (s_t, g^\star, g_t, r_t^H, s_{t+k}), \tag{1}$$

is relabeled as:

$$\widetilde{\Sigma}_t = (s_t, g^\star, g_t, r_t^\varphi, s_{t+k}), \tag{2}$$

where the *language-guided reward* is defined as $r_t^\varphi := r_\varphi(s_t, g^\star, g_t)$.

Since $r_\varphi$ does not depend on the lower-level policy $\pi_L$ or the agent's trajectories, it remains *stationary* across training iterations. This removes reward drift prevalent in off-policy HRL, thereby stabilizing the learning process for the higher-level policy. Consequently, the relabeled buffer $\mathcal{D}_H = \{\widetilde{\Sigma}_t\}$ is used for training the higher-level policy.

## 4.3 ADDRESSING REWARD SPARSITY WITH HER

In environments with sparse rewards, the high-level reward signal, $r_\varphi$, can often can be too sparse to produce a meaningful reward signal. To overcome this challenge, **LGR2** incorporates goal-conditioned hindsight experience replay (Andrychowicz et al., 2017) to densify the reward signal.

Concretely, for each relabeled high-level transition $\tilde{\Sigma}_t$, we sample alternative goals $\hat{g}$ from states encountered within the same trajectory, i.e.,

$$\hat{g} \in \{s_{t+j} \mid 0 \leq j \leq k\}.$$

We then relabel the transition as

$$\tilde{\Sigma}_t^{\text{HER}} = \left(s_t, \hat{g}, g_t, r_\varphi(s_t, \hat{g}, g_t), s_{t+k}\right),$$

which is added to the replay buffer $\mathcal{D}^H$. This hindsight relabeling increases the likelihood of encountering positive rewards in the replay buffer, thereby enhancing sample efficiency. The boosted frequency of informative reward signals encourages more effective exploration and leads to improved training stability in sparse-reward, long-horizon tasks.

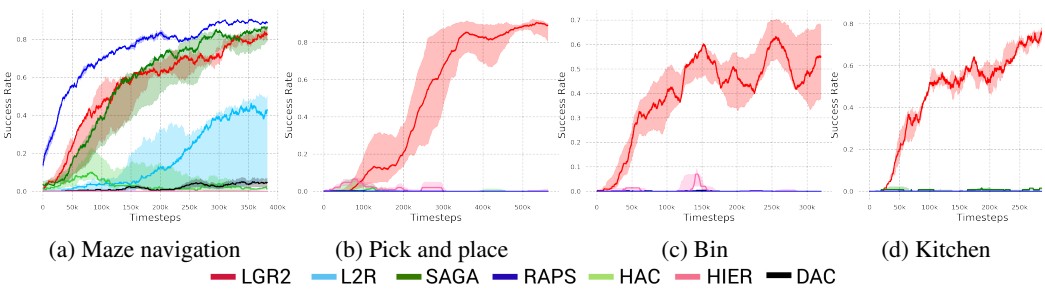

|     |     |     |     |
| --- | --- | --- | --- |
| (a) Maze navigation | (b) Pick and place | (c) Bin | (d) Kitchen |

LGR2 ▬ L2R ▬ SAGA ▬ RAPS ▬ HAC ▬ HIER ▬ DAC

Figure 2: **Success rate comparison** This figure compares the success rate performances on four sparse maze navigation and robotic manipulation environments. The solid line and shaded regions represent the mean and standard deviation, across 5 seeds. We compare our approach LGR2 against multiple baselines. LGR2 shows impressive performance and significantly outperforms the baselines.

## 4.4 TRAINING OBJECTIVES

Both the lower-level and higher-level policies are trained using the Soft Actor-Critic Haarnoja et al. (2018) framework with separate critic networks, adapted to their respective roles in the hierarchical setup.

**Lower-Level Policy Loss** The lower-level policy $\pi_L$ aims to achieve the subgoals issued by the higher-level policy within $k$ environment steps. Its training relies on a standard SAC critic update based on the replay buffer $\mathcal{D}_L$:

$$J_L = \mathbb{E}_{(s,g,a,r,s')\sim\mathcal{D}_L}\left[\left(r + \gamma Q^L_{\bar{\theta}}(s',g) - Q^L_{\theta}(s,g,a)\right)^2\right], \tag{3}$$

where $s, s' \in \mathcal{S}$ are states, $g \in \mathcal{G}$ is the current goal or subgoal, $a \in \mathcal{A}$ is the action, $r$ is the reward signal for the lower-level policy, $Q^L_{\theta}$ is the current lower-level critic parameterized by $\theta$, and $Q^L_{\bar{\theta}}$ is the target critic network.

**Higher-Level Policy Loss** The higher-level policy $\pi_H$ is trained using relabeled, stationary rewards derived from language-guided reward functions ($r_{\varphi}$) and the higher-level replay buffer $\tilde{\mathcal{D}}_H$:

$$J_H = \mathbb{E}_{(s,g^\star,g,r_{\varphi},s')\sim\tilde{\mathcal{D}}_H}\left[\left(r_{\varphi} + \gamma^k Q^H_{\bar{\omega}}(s',g') - Q^H_{\omega}(s,g^\star,g)\right)^2\right], \tag{4}$$

where $s, s' \in \mathcal{S}$, $g^\star \in \mathcal{G}$ is the final goal, $g \in \mathcal{G}$ is the current subgoal, $r_{\varphi}$ is the stationary, language-guided reward, $Q^H_{\omega}$ and $Q^H_{\bar{\omega}}$ are the current and target higher-level critics parameterized by $\omega$ and $\bar{\omega}$ respectively, and $g' \sim \pi_H(s')$ is the next subgoal sampled from the higher-level policy.

The language-guided reward $r_{\varphi}$ is invariant with respect to the evolving lower-level policy, thus enhancing training stability. We provide the algorithm in Supplementary Section 2.

## 5 EXPERIMENTS

In this section, we investigate the following questions:
1. How does LGR2 perform on complex, sparse-reward robotic navigation and manipulation tasks compared to prior hierarchical and single-level baselines?
3. Can language-guided reward relabeling generate better rewards than alternative baselines?
4. Does language-guided reward relabeling outperform standard HRL with hindsight relabeling?
5. Can LGR2 mitigate non-stationarity in HRL?
6. What is the contribution of hindsight experience relabeling (HER) in improving performance?
7. Can LGR2 policies transfer to real-world robots?

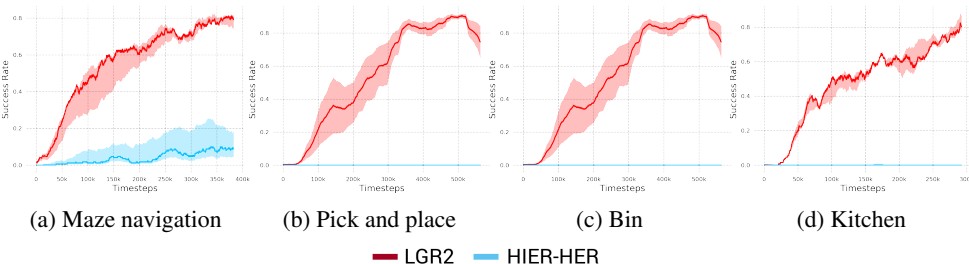

(a) Maze navigation    (b) Pick and place    (c) Bin    (d) Kitchen

— LGR2 — HIER-HER

Figure 3: **Comparison against HRL+HER** This figure compares the cumulative rewards on four sparse maze navigation and robotic manipulation environments. We compare our approach LGR2 against the hierarchical baseline HIER-HER. LGR2 outperforms this baseline, which shows the efficacy of language-guided reward generation in LGR2.

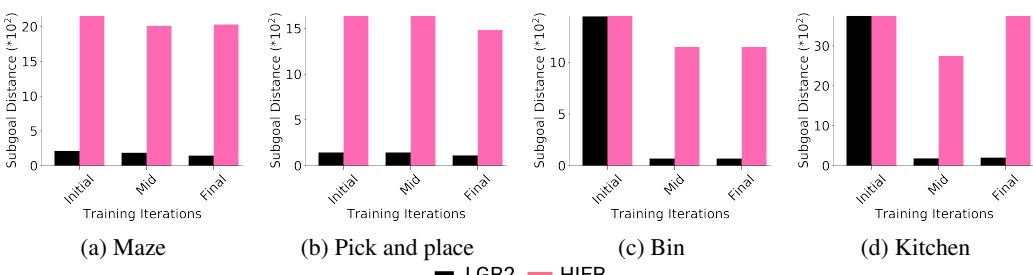

(a) Maze    (b) Pick and place    (c) Bin    (d) Kitchen

— LGR2 — HIER

Figure 4: **Non-stationarity metric comparison** This figure compares the average distance metric between the subgoals predicted by the higher level policy and the states achieved by the lower level policy during training. (The columns represent Initial: when training begins, Mid: half-way during training, Final: when training ends, e.g. since maze navigation is trained for 3.8$\mathbf{E}$5 timesteps, the values are Initial: iteration 1, Mid: iteration 1.9$\mathbf{E}$5, and Final: iteration 3.8$\mathbf{E}$5). As can be seen, LGR2 consistently generates efficient and achievable subgoals, thereby mitigating non-stationarity in HRL.

**Implementation details.** We evaluate LGR2 on four continuous sparse reward robotic tasks: maze navigation, pick and place, bin, and franka kitchen (Gupta et al., 2019). In the maze task, intermediate waypoints serve as subgoals; for more complex tasks like franka kitchen, full intermediate states are used as subgoals. The appendix, implementation and environment details (supplementary Sections 4 and 6), additional hyper-parameters (supplementary Sections 5), code, and a qualitative video are provided in the supplementary. The policy networks consist of three fully connected layers with 512 units each, trained using SAC (Haarnoja et al., 2018) and Adam optimizer (Kingma & Ba, 2014). Refer to the supplementary for motion descriptor (Sec. 7), reward generator (Sec. 7), sample codes (Sec. 8), and qualitative visualizations (Sec. 10). We carefully tune hyperparameters via grid search across all baselines to ensure fair comparison. For challenging tasks such as pick and place and kitchen, we incorporate a single demonstration and an imitation learning objective at the lower level; no demonstrations are used for maze navigation to maintain consistent evaluation.

**Sparse Rewards and Challenges of Goal-Conditioning.** Although some tasks may appear simple, all our evaluation environments feature inherently sparse reward structures, requiring agents to engage in extensive exploration before encountering any positive feedback. This sparse reward setting significantly amplifies the difficulty of the tasks. Additionally, we adopt a goal-conditioned RL framework with randomly generated initial positions and final goals, further increasing the complexity by demanding generalization across diverse objectives. These factors combined lead to markedly challenging training conditions, which explain why most baseline methods perform poorly in our experiments. The sparse and delayed nature of the rewards, coupled with high variability in goals, makes efficient credit assignment and exploration extremely difficult for flat or naive hierarchical policies. In contrast, our hierarchical approach benefits from temporal abstraction, and is able to significantly outperform the baselines in most tasks.

**Computational Overhead of LLM-Integrated Reward Generation.** While LGR2 leverages LLMs for expressive, flexible reward generation, the main computational cost, ie translating natural language

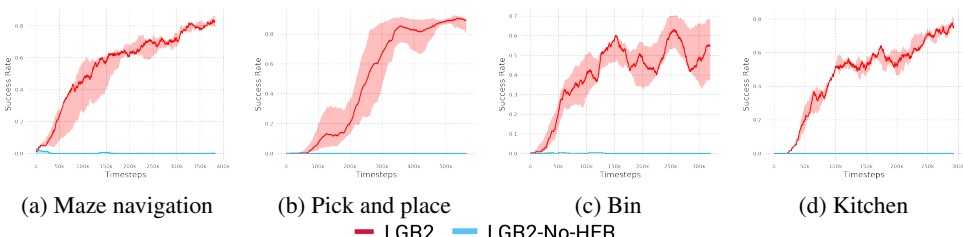

(a) Maze navigation     (b) Pick and place     (c) Bin     (d) Kitchen

LGR2     LGR2-No-HER

Figure 5: **Hindsight Relabeling ablation** This figure compares the performance of LGR2 with LGR2-No-HER ablation (LGR2 without hindsight relabeling). The plots clearly demonstrate that HER is crucial for good performance in all the tasks.

instructions into structured motion descriptors and reward-coder parameters via LLM calls, is incurred *once per task specification*, before interaction and training. This offline design prevents LLM calls from bottlenecking RL, as later reward evaluations are lightweight, involving only fast code execution. Memory impact is minimal since only the reward code or parameters are stored, not full LLM outputs. LGR2 maintains sample efficiency and is practical for both simulation and real robotic applications.

**How does LGR2 perform on complex control tasks compared to prior baselines?**
We compare LGR2 with prior baselines (Figure 2); curves show mean ± std over 5 seeds.
**L2R.** The L2R approach (Yu et al., 2023) translates natural language instructions into reward parameters via a reward translator and originally uses an MPC controller (Howell et al., 2022). For fair comparison, our baseline adopts the same translator but replaces MPC with a Soft Actor-Critic (SAC) (Haarnoja et al., 2018) agent. While this single-level setup achieves some progress in maze tasks, it struggles in harder sparse-reward domains, whereas LGR2 consistently outperforms L2R by leveraging hierarchical structure for temporal abstraction and subgoal decomposition (see Figure 2).
**SAGA.** We compare LGR2 with SAGA (Wang et al., 2023), a hierarchical method that uses a state-conditioned discriminator to align subgoals with the low-level policy. While effective on maze tasks, SAGA deteriorates on harder benchmarks, showing susceptibility to instability in long-horizon settings. In contrast, LGR2 achieves stronger results by leveraging language-guided reward relabeling, which provides a stationary, semantically meaningful signal that stabilizes hierarchical training.

**RAPS.** The RAPS baseline (Dalal et al., 2021) uses a library of predefined robot action primitives controlled by the higher-level policy. While RAPS performs well on simpler maze tasks, it struggles on all other harder tasks. This suggests methods relying on fixed primitives lack flexibility for complex objectives. In contrast, LGR2 autonomously discovers effective subgoal decompositions and adapts dynamically, achieving strong results on multi-step, long-horizon tasks.
**HAC.** HAC (Andrychowicz et al., 2017) attempts to mitigate non-stationarity by assuming an optimal lower-level policy and enhances the higher-level replay buffer using subgoal relabeling. Despite this, our experiments show LGR2 surpasses HAC (see Figure 2). We believe that this is because LGR2's language-guided reward relabeling generates stable, goal-aligned rewards for the higher-level policy without requiring the unrealistic assumption of an optimal lower-level primitive, thus providing a more practical and robust mechanism for addressing non-stationarity. HIRO (Nachum et al., 2018) is another such baseline that addresses non-stationarity, however since HAC has been found to outperform HIRO, we compared with HAC baseline.
**HIER.** The HIER baseline is a standard hierarchical SAC model where the higher-level reward is the cumulative sum of environment rewards across $k$ steps. LGR2 achieves much higher performance than HIER, underscoring the critical role of language-guided reward generation in reducing non-stationarity and resulting in stable hierarchical learning.
**DAC.** Finally, we also consider DAC baseline with access to a single demonstration. DAC fails to show any significant progress, as seen in Figure 2, which shows that LGR2 outperforms single-level baselines with access to privileged information like expert demonstrations.

**Can language-guided reward relabeling achieve better rewards than alternative baselines?**
To evaluate the effectiveness of language-guided reward relabeling in densifying rewards and promoting exploration, we compare the *average reward per episode* achieved by LGR2 with two hierarchical baselines: HAC and HIER. As shown in Supplementary Sec. 3 Figure 1, LGR2 consistently achieves substantially higher average rewards throughout training, demonstrating its ability to generate richer

rewards. This reflects the benefit of using language-derived, relabeled high-level rewards, which provide semantically meaningful feedback even in highly sparse environments. In contrast, HAC and HIER both suffer from limited reward density, leading to slower learning and inefficient exploration. These results underscore that our language-guided reward relabeling is key to producing informative rewards and improve performance in complex, sparse-reward robotic tasks.

**Does language-guided reward relabeling outperform standard HRL with hindsight relabeling?**
We assess the isolated contribution of language-guided reward relabeling by comparing LGR2 with the ablation *HIER-HER*, a hierarchical RL baseline that uses HER but relies solely on environment signals for high-level rewards, with no language guidance. This setup retains all elements of LGR2 except language-driven reward relabeling. As shown in Figure 3, LGR2 consistently outperforms HIER-HER across all tasks, achieving greater learning efficiency and higher success rates in complex sparse-reward environments. These improvements highlight that while HER effectively densifies feedback, the core gains of LGR2 stem from the ability of language-guided relabeling to provide a stationary, semantically aligned supervisory signal for the high-level policy. This stabilizes hierarchical learning and leads to improved performance over baselines.

**Can LGR2 mitigate non-stationarity in HRL?**
We show LGR2's ability to mitigate non-stationarity in HRL in Figure 4. To quantify this, we compare LGR2 to the *HIER* baseline by measuring the average distance between subgoals predicted by the higher-level policy and the final states reached by the lower-level policy at different training stages (Initial: start of training, Mid: halfway point, Final: end of training). Lower distance values indicate that the higher-level policy generates subgoals that are feasible and well-aligned with the abilities of the evolving lower-level policy, leading to more consistent goal achievement and reduced non-stationarity. As shown in Figure 4, LGR2 consistently achieves lower average distances across training, validating that it produces achievable subgoals and robustly reduces non-stationarity compared to the baseline.

**What is the contribution of hindsight experience relabeling (HER) in improving performance?**
Further, we analyse the importance of hindsight experience replay (HER) (Andrychowicz et al., 2017) for densifying rewards in sparse reward tasks. To this end, we compare LGR2 with LGR2-No-HER baseline, which is LGR2 baseline without HER. As seen in Figure 5, HER is crucial for good performance, since language-guided rewards are too sparse to generate any meaningful reward signal.

**Can LGR2 policies transfer to real-world robots?**
We conducted real-world experiments on pick-and-place and bin tasks (supplementary Section 9), using a RealSense D435 depth camera to track the robotic arm, block, and bin positions. The robot was controlled via the manufacturer's Python SDK at 20 Hz ($\Delta t = 50$ ms), with episodes lasting about 40 seconds. Policies were trained in simulation and then deployed on the robot; due to challenges in precise velocity control, we used small fixed velocities, achieving good performance. Across 5 sets of 10 trials each, LGR2 attained average success rates of 60% and 50%, with variances 0.07 and 0.03 for pick-and-place and bin tasks, respectively. The best baseline, L2R, failed to show significant progress. The supplementary also video depicts real-world evaluation.

## 6 CONCLUSION

**Limitations and Future Work.** While LGR2 mitigates reward-level non-stationarity through language-guided relabeling, it does not address transition dynamics shifts from the evolving lower-level policy. Future work could extend LGR2 to handle transition non-stationarity via predictive models and automate prompt generation to reduce manual design effort and potential hallucinations.
**Discussion.** In this work, we introduced **LGR2**, a novel HRL framework that leverages language-guided reward relabeling to address non-stationarity by deriving invariant reward functions from natural language instructions. Combined with HER for reward densification, LGR2 enables efficient learning on complex, long-horizon tasks without requiring extensive demonstrations. Our approach consistently outperforms strong baselines across challenging simulated and real-world robotic tasks, demonstrating the effectiveness of integrating large language models into HRL pipelines. This work underscores the importance of harnessing semantic knowledge from natural language for scalable robotic control. LGR2 represents a promising step toward robots that robustly interpret and execute human instructions, with future research expected to further broaden the scope of language-conditioned hierarchical policies.

## ETHICS STATEMENT

This work presents LGR2, a hierarchical reinforcement learning framework that integrates large language models for generating reward functions in robotic navigation and manipulation tasks. We recognize several ethical considerations associated with this research. First, our method relies on LLMs to translate natural language instructions into reward parameters, which may inherit biases from the training data of these models, potentially leading to unintended or unfair behaviors in robotic systems, especially in diverse real-world applications. Second, the focus on robotic tasks such as pick-and-place, bin manipulation, and kitchen environments could contribute to advancements in automation, which may impact employment in manual labor sectors. While LGR2 aims to enhance efficiency and stability in sparse-reward settings, we emphasize the need for careful assessment of its societal effects, including equitable access to such technologies. Third, training hierarchical policies requires substantial computational resources, contributing to energy consumption and environmental footprint; we advocate for optimized implementations to minimize this impact. Fourth, all datasets and environments used (e.g., maze navigation, Franka kitchen from D4RL) comply with their respective licenses, and our approach does not introduce new privacy risks beyond standard RL practices. We encourage ethical deployment with emphasis on transparency, bias mitigation, and human oversight.

## REPRODUCIBILITY STATEMENT

To facilitate reproducibility, we provide detailed documentation of our methods and experiments. The full mathematical formulation of the language-guided reward generation, relabeling process, and training objectives (including SAC adaptations) is presented in Section 4, with pseudocode in Algorithm 1 and additional details in Appendix A.4. Hyperparameter configurations, such as learning rates, batch sizes, network architectures (three fully connected layers with 512 units), and SAC parameters (e.g., alpha=0.05, tau=0.8), are specified in Appendix Table 1. Experiments span four robotic environments (maze navigation, pick-and-place, bin, Franka kitchen), with configurations, state/action spaces, reward structures, and evaluation metrics (e.g., success rates, non-stationarity distance) described in Section 5 and Appendix A.4. Prompts for LLM-based motion descriptors and reward coders are fully provided in Appendix A.5, along with sample generated codes in Appendix A.5. Baseline implementations (e.g., L2R, SAGA, RAPS, HAC, HIER, DAC) are detailed in Section 5 for fair comparisons, including tuning via grid search. We intend to release the complete codebase, including LLM integration, replay buffer management, and evaluation scripts, upon publication. Statistical analyses, such as means and standard deviations over 5 seeds, are reported with confidence intervals where applicable. Computational requirements (e.g., Intel Core i7, NVIDIA GTX 1080, training timesteps) are outlined in Appendix A.3 to support replication on similar hardware.

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

CONTENTS

## A   APPENDIX

### A.1   BROADER IMPACT STATEMENT

Our proposed approach and algorithm are not aimed at delivering immediate technological break-throughs. Instead, our work makes a conceptual contribution by addressing fundamental aspects of Hierarchical Reinforcement Learning (HRL). By leveraging language-guided instructions, we establish a novel framework with the potential to significantly propel research in HRL and related areas. This conceptual foundation lays the groundwork for future studies and may catalyze further advancements in the field.

### A.2   LGR2 ALGORITHM

The detailed LGR2 pseudocode is provided here:

---

**Algorithm 1** LGR2

---

1: Initialize higher level replay buffer $\mathcal{D}^H = \{\}$ and lower level replay buffer $\mathcal{D}^L = \{\}$
2: **for** $i = 1 \ldots N$ **do**
3:      // Collect transitions using $\pi^H$ and store in $\mathcal{D}^H$
4:      // Collect transitions using $\pi^L$ and store in $\mathcal{D}^L$
5:      **for** each timestep $t$ **do**
6:          $d^H \leftarrow d^H \cup \{(s_t, g^*, g_t, r_t^H, s_{t+k-1})\}$
7:          $d^L \leftarrow d^L \cup \{(s_t, g_t, a_t, r_t^L, s_{t+1})\}$
8:      $\mathcal{D}^H \leftarrow \mathcal{D}^H \cup d^H$
9:      $\mathcal{D}^L \leftarrow \mathcal{D}^L \cup d^L$
10:      // Sample and relabel higher-level trajectories
11:      **for** $i = 1 \ldots M$ **do**
12:          $\sigma = \{(s_t, g^*, g_t, r_t^H, s_{t+k-1})\}_{t=1}^{n-1} \sim \mathcal{D}^H$
13:          // Relabel the reward by language-guided $r_\phi$
14:          // Store the transition in $\mathcal{D}^H$
15:          $\mathcal{D}^H \leftarrow \mathcal{D}^H \cup \{(s_t, g^*, g_t, r_\phi, s_{t+k-1})\}_{t=1}^{n-1}$
16:      // Sample a set of additional goals for HER ($\widehat{G}$)
17:      **for** $\widehat{g} \in \widehat{G}$ **do**
18:          // Relabel $g$ by $\widehat{g}$ and $r_\phi$ by $\widehat{r}_\phi$ in $\sigma$ such that
19:          $\widehat{\sigma} = \{(s_t, \widehat{g}, g_t, \widehat{r}_\phi, s_{t+k-1})\}_{t=1}^{n-1}$
20:          Store in replay buffer $\mathcal{D}^H \leftarrow \mathcal{D}^H \cup \widehat{\sigma}$
21:      // Policy Learning
22:      **for** each gradient step **do**
23:          Sample $\{(\sigma_j)\}_{j=1}^m$ from $\mathcal{D}^H$
24:          Sample $\{(\delta_j)\}_{j=1}^m$ from $\mathcal{D}^L$
25:          Optimize higher policy $\pi^H$ using SAC
26:          Optimize lower policy $\pi^L$ using SAC

---

### A.3   IMPLEMENTATION DETAILS

We conduct our experiments on two systems, each equipped with an Intel Core i7 processor, 48GB of RAM, and an Nvidia GeForce GTX 1080 GPU. We also report the number of timesteps required for running the experiments.

In our setup, both the actor and critic networks are implemented as three-layer, fully connected neural networks, each with 512 neurons per layer.

For the maze navigation task, a 7-degree-of-freedom (7-DoF) robotic arm moves through a four-room maze with its closed gripper fixed at table height, navigating to reach the goal position. In the pick-and-place task, the same 7-DoF robotic arm identifies a square block, picks it up, and delivers it to the goal position. In the bin environment, the gripper must pick up the block and place it in a designated bin. Lastly, in the kitchen task, a 9-DoF Franka robot performs a predefined complex action—opening a microwave door—to complete the task.

To ensure fair comparisons, we maintain consistency across all baselines by keeping key parameters unchanged wherever possible. These include the neural network layer width, the number of layers, the choice of optimizer, and the SAC implementation parameters.

We also provide the hyperparameter configuration in Table 1

### A.4   ENVIRONMENT DETAILS

In this section, we provide the environment and implementation details for all the tasks:

Table 1: Hyperparameter Configuration

| Parameter | Value | Description |
|---|---|---|
| activation | tanh | activation for hierarchical policies |
| layers | 3 | number of layers in the critic/actor networks |
| hidden | 512 | number of neurons in each hidden layer |
| Q_lr | 0.001 | critic learning rate |
| pi_lr | 0.001 | actor learning rate |
| buffer_size | int(1E7) | for experience replay |
| tau | 0.8 | polyak averaging coefficient |
| clip_obs | 200 | clip observation |
| n_cycles | 1 | per epoch |
| n_batches | 10 | training batches per cycle |
| batch_size | 1024 | batch size hyper-parameter |
| random_eps | 0.2 | percentage of time a random action is taken |
| alpha | 0.05 | weightage parameter for SAC |
| noise_eps | 0.05 | std of gaussian noise added to not-completely-random actions |
| norm_eps | 0.01 | epsilon used for observation normalization |
| norm_clip | 5 | normalized observations are cropped to this value |
| adam_beta1 | 0.9 | beta 1 for Adam optimizer |
| adam_beta2 | 0.999 | beta 2 for Adam optimizer |

### A.4.1 MAZE NAVIGATION ENVIRONMENT

In this environment, a 7-DOF robotic arm gripper navigates through randomly generated four-room mazes to reach the goal position. The gripper remains closed and fixed at table height, with the positions of walls and gates randomly determined. The table is divided into a rectangular $W \times H$ grid, and the vertical and horizontal wall positions, $W_P$ and $H_P$, are randomly selected from $(1, W - 2)$ and $(1, H - 2)$, respectively. In the constructed four-room environment, the four gate positions are randomly chosen from $(1, W_P - 1)$, $(W_P + 1, W - 2)$, $(1, H_P - 1)$, and $(H_P + 1, H - 2)$.

In the maze environment, the state is represented as the vector $[dx, M]$, where $dx$ denotes the current gripper position and $M$ is the sparse maze array. The higher-level policy input is a concatenated vector $[dx, M, g]$, where $g$ is the target goal position. The lower-level policy input is a concatenated vector $[dx, M, s_g]$, where $s_g$ is the sub-goal provided by the higher-level policy. $M$ is a discrete 2D one-hot vector array, with 1 indicating the presence of a wall block. The lower primitive action $a$ is a 4-dimensional vector, with each dimension $a_i \in [0, 1]$. The first three dimensions provide offsets to be scaled and added to the gripper position for movement. The last dimension controls the gripper, with 0 indicating a closed gripper and 1 indicating an open gripper.

### A.4.2 PICK AND PLACE AND BIN ENVIRONMENTS

In this section, we describe the environment details for the pick and place and bin tasks. The state is represented as the vector $[dx, o, q, e]$, where $dx$ is the current gripper position, $o$ is the position of the block object on the table, $q$ is the relative position of the block with respect to the gripper, and $e$ includes the linear and angular velocities of both the gripper and the block object. The higher-level policy input is a concatenated vector $[dx, o, q, e, g]$, where $g$ is the target goal position. The lower-level policy input is a concatenated vector $[dx, o, q, e, s_g]$, where $s_g$ is the sub-goal provided by the higher-level policy. In our experiments, the sizes of $dx$, $o$, $q$, and $e$ are set to 3, 3, 3, and 11, respectively. The lower primitive action $a$ is a 4-dimensional vector with each dimension $a_i \in [0, 1]$. The first three dimensions provide gripper position offsets, and the last dimension controls the gripper. During training, the positions of the block object and the goal are randomly generated (the block is always initialized on the table, and the goal is always above the table at a fixed height).

### A.4.3 FRANKA KITCHEN ENVIRONMENT

For this environment please refer to the D4RL environment Fu et al. (2020). In this environment, the franka robot has to perform a complex multi-stage task in order to achieve the final goal.

## A.5    FULL PROMPTS

In this section, we provide detailed prompts for motion description and reward translator for all the environments.

### A.4.1. Maze Navigation Environment

We now present the motion descriptor and reward generator prompts for the maze navigation environment.

**Motion Descriptor Prompt for Maze Navigation environment**

We want you to generate a random position for an object within the table following the description and rules.

[Description]

1. There is a table which can be represented as a matrix of (num_1, num_2).

2. Generate walls within the table by choosing a random row and random column and blocking all of (num_1+num_2) cells.

3. Generate four random cells from the (num_1+num_2) marked as walls. Remove the blocks from these cells and mark them as gates.

4. Generate final position for the object like CHOICE:[cuboid,apple,ball] with the (x,y) co-ordinates between (num_4,num_5) and height is at table height in the bottom right room and bottom right corner.

Rules

1. The robot is a 7-DOF robotic arm gripper.

2. The height of the table is table_height=0.42 cm.

3. If you see num_1 replace it with an integer within 10 and 20.

4. If you see num_2 replace it with an integer within 10 and 20.

5. If you see phrases like CHOICE: [choice1, choice2, ...], it means you should replace the entire phrase with one of the choices listed.

6. Please remember that the final position cannot coincide with gates, walls or starting position location.

7. The starting position of the location is (1,3)

**Reward Generator Prompt for Maze Navigation maze environment**

We have a description of a robot's motion and we want you to turn that into the corresponding program with following functions:

def reset_environment()

def set_Gripper_Pos(x_pos, y_pos, z_pos)

x_pos: position of x-coordinate of the gripper of robot arm.

y_pos: position of y-coordinate of the gripper of robot arm.

z_pos: position of z-coordinate (height) of the gripper of robot arm.

generate_maze()

```
do_simulation()

Example answer code:

import numpy as np reset_environment()

# This is a new task so reset environment else we do not need it.

set_Gripper_Pos(3.0,2.0,0.56)

set_Gripper_Pos(4.45,3.56,0.48)

set_Gripper_Pos(6.85,7.36,0.64)

generate_maze()

# generate maze with all the constraints

do_simulation()

# run the simulation
```

### A.4.2. Pick and Place Environment

We now present the motion descriptor and reward generator prompts for the pick and place environment.

**Motion Descriptor Prompt for Pick and Place Environment**

We want you to generate a random position for an object within the table following the description and rules.

[ Description]

1. There is a table which can be represented as a matrix of (num_1, num_2) and height (num_3).

2. Generate final position for the object like CHOICE:[cuboid,apple,ball] with the (x,y) co-ordinates between (num_4,num_5) and height (num_6).

Rules:

1. The robot is a 7-DOF robotic arm gripper.

2. The height of the table is table_height=0.42 cm.

3. The max height the arm can reach is max_height=0.66 cm.

4. If you see num_1 replace it with an integer within 10 and 20.

5. If you see num_2 replace it with an integer within 10 and 20.

6. If you see num_3 replace it with 0.42.

7. If you see phrases like CHOICE: [choice1, choice2, ...], it means you should replace the entire phrase with one of the choices listed.

8. Please remember that there is an object on the table.

9. The block is light enough for the robot to pick up and hold in the air for a long time, like 4 seconds.

**Reward Generator Prompt for Pick and Place Environment**

We have a description of a robot's motion and we want you to turn that into the corresponding program with following functions:

def reset_environment()

def set_Gripper_Pos(x_pos, y_pos, z_pos)

x_pos: position of x-coordinate of the gripper of robot arm.

y_pos: position of y-coordinate of the gripper of robot arm.

z_pos: position of z-coordinate (height) of the gripper of robot arm.

def generate_Object_Pos()

def do_simulation()

Example answer code:

import numpy as np

reset_environment()

# This is a new task so reset environment else we do not need it.

set_Gripper_Pos(3.0,2.0,0.56)

set_Gripper_Pos(4.45,3.56,0.48)

set_Gripper_Pos(6.85,7.36,0.64)

do_simulation()

# run the simulation

### A.4.3. Bin Environment

We now present the motion descriptor and reward generator prompts for the bin environment.

**Motion descriptor prompt for Bin environment**

We want you to generate a random position for a bin and an object within the bin following the description and rules. [Description]

1. There is a table which can be represented as a matrix of (num_1, num_2) and height (num_3).
2. There is a bin on the table.
3. Generate a random position for the bin within the table.
4. Generate a final position (x,y) for placing the object like CHOICE:[cuboid,apple,ball] within the bin

[Rules]

1. The robot is a 7-DOF robotic arm gripper.
2. The height of the table is table_height=0.42 cm.
3. The max height the arm can reach is max_height=0.66 cm.
4. If you see num_1 replace it with an integer within 10 and 20.
5. If you see num_2 replace it with an integer within 10 and 20.

6. If you see num_3 replace it with 0.42.

7. The bin has to be completely within the table. No part of the bin can be outside of the table.

8. The height, width and length of the table are 0.1 cm, respectively.

9. The final position of the object should be continuous and at the centre of the bin.

10. If you see phrases like CHOICE: [choice1, choice2, ...], then you should be replacing the entire phrase with one of the choices listed.

11. Please remember that there is always a bin on the table.

12. The object is light enough for the robot to pick up and hold in the air for a long time, like 4 seconds.

**Reward generator prompt for Bin environment**

We have a description of a robot's motion and we want you to turn that into the corresponding program with following functions:

def reset_environment()

def set_Gripper_Pos(x_pos, y_pos, z_pos)

x_pos: position of x-coordinate of the gripper of robot arm.

y_pos: position of y-coordinate of the gripper of robot arm.

z_pos: position of z-coordinate (height) of the gripper of robot arm.

def generate_Bin()

def do_simulation()

Example answer code:

import numpy as np

reset_environment()

# This is a new task so reset environment else we do not need it.

set_Gripper_Pos(3.0,2.0,0.56)

set_Gripper_Pos(4.45,3.56,0.48)

set_Gripper_Pos(6.85,7.36,0.64)

generate_Bin()

# generate bin with all the constraints

do_simulation()

# run the simulation

### A.4.4. Franka Kitchen Environment

We now present the motion descriptor and reward generator prompts for the franka kitchen environment.

**Motion descriptor prompt for Franka kitchen environment**

We want you to generate a random position for the door of a microwave and gas-knob following the description and rules.

[Description]

1. There is a microwave and gas knob.
2. Open the microwave door to a certain point num_1.
3. Turn the gas knob to co-ordinates (num_2, num_3).

[Rules]

1. The robot is a 7-DOF robotic arm gripper.
2. The height of the table is table_height=0.42 cm.
3. The max height the arm can reach is max_height=0.66 cm.
4. If you see num_1 replace it with a continuous number -0.75.
5. If you see num_2 replace it with a continuous number -0.88.
6. If you see num_2 replace it with a continuous number -0.01.

**Reward generator prompt for Franka kitchen environment**

We have a description of a robot's motion and we want you to turn that into the corresponding program with following functions:

def reset_environment()

def set_microwave_Pos(joint_1_Pos)

joint_1_Pos: Position of joint to determine whether microwave door is open or not.

def set_gas_Knob_Pos(joint_1_Pos, joint_2_Pos)

joint_1_Pos: x co-ordinate of the gas knob.

joint_2_pos: y co-ordinate of the gas knob.

def do_simulation()

Example answer code:

import numpy as np

reset_environment()

# This is a new task so reset environment else we do not need it.

set_microwave_Pos(-0.75)

set_microwave_Pos(-0.55)

set_gas_Knob_Pos(-0.88, -0.01)

set_gas_Knob_Pos(-0.75, -0.1)

do_simulation()

# run the simulation

## A.6 SAMPLE CODES

In this section, we show sampled responses for all four environments mentioned in this paper for a better understanding of the LGR2 method. We sample 2-3 responses for each method to show both successful and failed results.

### A.5.1. Maze Navigation Environment

**Sample Code for Maze Navigation environment**

Here, we provide the sample code for the Maze Navigation environment generated by GPT-3.5.

```python
import numpy as np
def reset_environment():
    print("Environment reset.")
def set_Gripper_Pos(x_pos, y_pos, z_pos):
    print(f"Setting gripper position to x: x_pos, y: y_pos, z: z_pos.")
def generate_maze():
print("Generating maze...")
def do_simulation():
    print("Running simulation...")
reset_environment()
# Set the gripper positions for linear motion
set_Gripper_Pos(3.0, 2.0, 0.56)
set_Gripper_Pos(4.45, 3.56, 0.48)
set_Gripper_Pos(6.85, 7.36, 0.64)
# Generate maze
generate_maze()
# Run simulation
do_simulation()
```

### A.5.2. Pick and Place Environment

**Sample Code for Pick and Place environment**

Here, we provide the sample code for the Pick and Place environment generated by GPT-3.5.

```python
import numpy as np
def reset_environment():
    print("Environment reset.")
def set_Gripper_Pos(x_pos, y_pos, z_pos):
    print(f"Setting gripper position to x: x_pos, y: y_pos, z: z_pos.")
def generate_Object_Pos(): print("Generating object position.")
```

```python
def do_simulation():
    print("Simulation started.")
reset_environment()
set_Gripper_Pos(3.0, 2.0, 0.56)
set_Gripper_Pos(4.45, 3.56, 0.48)
set_Gripper_Pos(6.85, 7.36, 0.64)
generate_Object_Pos()
do_simulation()
```

### A.5.3. Bin Environment

**Sample Code for Bin environment**

Here, we provide the sample code for the Bin environment generated by GPT-3.5.

```python
import numpy as np
def reset_environment():
    print("Environment reset.")
def set_Gripper_Pos(x_pos, y_pos, z_pos):
    print(f"Setting gripper position to x: x_pos, y: y_pos, z: z_pos.")
def generate_Bin():
    print("Generating bin with all constraints.")
def do_simulation():
    print("Simulation started.")
reset_environment()
set_Gripper_Pos(3.0, 2.0, 0.56)
set_Gripper_Pos(4.45, 3.56, 0.48)
set_Gripper_Pos(6.85, 7.36, 0.64)
generate_Bin()
do_simulation()
```

### A.5.4. Franka Kitchen Environment

**Sample Code for Franka kitchen environment**

Here, we provide the sample code for the Franka kitchen environment generated by GPT-3.5.

```python
import numpy as np
def reset_environment():
    print("Environment reset")
def set_microwave_Pos(joint_1_Pos):
    print(f"Setting microwave door position to joint_1_Pos.")
```

```python
def set_gas_Knob_Pos(joint_1_Pos, joint_2_Pos):
    print(f"Setting gas knob position to x: joint_1_Pos, y: joint_2_Pos.")
def do_simulation():
    print("Simulation started.")
reset_environment()
set_microwave_Pos(-0.75)
set_microwave_Pos(-0.55)
set_gas_Knob_Pos(-0.88, -0.01)
set_gas_Knob_Pos(-0.75, -0.1)
do_simulation()
```

## A.7 QUALITATIVE VISUALIZATIONS

We provide qualitative visualizations for all the environments:

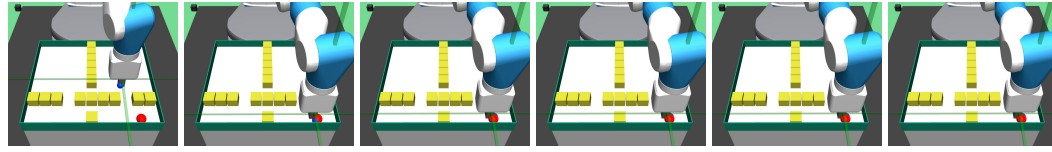

Figure 6: **Successful visualization**: The visualization is a successful attempt at performing maze navigation task

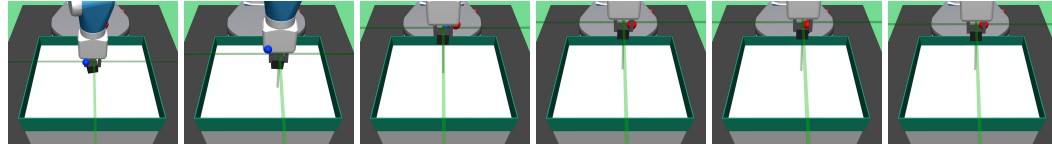

Figure 7: **Successful visualization**: The visualization is a successful attempt at performing pick and place task.

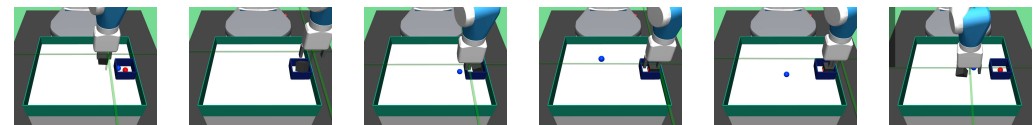

Figure 8: **Successful visualization**: The visualization is a successful attempt at performing bin task.

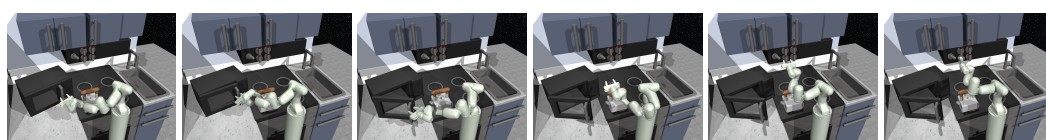

Figure 9: **Successful visualization**: The visualization is a successful attempt at performing kitchen task.

