# OpenReview forum: "LGR2: Language Guided Reward Relabeling for Accelerating Hierarchical Reinforcement Learning"
_ICLR.cc/2026/Conference — ICLR 2026 Conference Withdrawn Submission_

### Official Review · Reviewer_7atv · 2025-10-28

**Soundness:** 1
**Presentation:** 1
**Contribution:** 1
**Rating:** 0
**Confidence:** 4

**Summary:**

LGR2 learns a hierarchical reinforcement policy. LGR2 addresses the challenge of non-stationarity from training lower and higher-level policies by decoupling high-level reward generation from low-level policy changes. LGR2 uses LLMs to generate reward functions, which can be used for hindsight relabeling. LGR2 primarily shows results on sim benchmarks, with one real-world task. Primary baselines are other hierarchical RL works.

**Strengths:**

1. Thorough related works section.

**Weaknesses:**

1. Missing description of what subgoals are for all environments. I found some information in the Appendix A.4.1 and A.4.2 but it is missing for Franka kitchen.
2. Tasks are too simple. Maybe tasks like OGBench, or tasks that are more long-horizon, would be better suited for this HRL setup.
3. Results are not convincing, due to lack of baselines and more complex tasks.
4. Writing is a bit confusing at times, and could be more clearly organized. For instance, the experiments section is a bit confusing, and could be better structured so it explains the premises, focuses on main experimental questions and setup, and defers more niche experiments into an ablations section.
5. I could not find the real-world experiment in the pdf, and I also did not find a supplementary button on the page. The real-world experiment as it currently exists is not convincing.
6. In general, I find this setup not particularly convincing or practical. A more practical robotics application would be generating more useful subgoals, like images (SUSIE), or learning a language-conditioned policy. To study RL in sim, more complex, long-horizon tasks would be needed.

**Questions:**

1. Missing non-hierarchical baselines. How necessary is this hierarchy for relatively simple tasks? There are many papers that explore using VLM rewards for language-conditioned RL agents, which is similar to this problem setting. How does LGR2 compare to this?
2. Nit: line 310 missing 2. The numbering skips that (1, 3, 4).
3. Nit: Numbering should be done with \enumerate, not just directly in text.

---

### Official Review · Reviewer_KUDr · 2025-10-30

**Soundness:** 2
**Presentation:** 3
**Contribution:** 2
**Rating:** 2
**Confidence:** 4

**Summary:**

LGR2 is a hierarchical reinforcement learning (HRL) framework that leverages large language models (LLMs) to generate language-guided reward functions for the higher-level policy. It mitigates the non-stationarity problem in HRL caused by the changing behavior of the lower-level policy by producing stationary, semantically meaningful high-level rewards independent of the low-level policy, enabling stable learning in sparse, long-horizon robotic tasks.

**Strengths:**

1.1 Motivation is clear and strong: the authors target sparse tasks and the HRL non-stationarity problem directly.

1.2 The experiments are strong with multiple seeds, good baseline coverage, and consistent gains.

1.3 Sim-to-real results are also nice to see (with fairly good results).

**Weaknesses:**

2.1 The claim of invariance appears insufficiently justified. Although r_φ is defined over (s_t, g*, g_t), the state s_t itself depends on the evolving low-level policy, implying an indirect dependence on π_L. In effect, the method transforms a sparse high-level signal into a denser, language-guided shaping signal, which likely enhances learning stability but does not constitute true invariance.

2.2 The paper references a sim-to-real video in the supplementary, but no supplementary material was provided.

2.3 Please include concrete examples of r_H vs r_φ in the appendix code section. It’s hard to see the practical difference without a worked example, and it would be beneficial to show the actual reward functions used during training, which would clarify why r_φ helps.

2.4 The main paper does not state which LLM was used. First in the Appendix are examples of GPT-3.5 shown. This should have been stated earlier as to which backbone was used.

2.5 Related Work is dated. Critical recent works on language-to-code and language-to-reward are missing (the latest citation is from 2023).

2.6 The Experiments section reads partly like Intro/RW due to long baseline descriptions.

**Questions:**

3.1 Can you explain and expand more for my concern regarding point 2.1?

3.2 Can you also provide, the additional r_H vs r_φ examples as requested in 2.3.

3.3 Can you clarify the sim-to-real supplementary video (where is the link; what exactly is shown)?

---

### Official Review · Reviewer_EtCA · 2025-11-05

**Soundness:** 3
**Presentation:** 1
**Contribution:** 2
**Rating:** 4
**Confidence:** 4

**Summary:**

This manuscript uses an LLM to relabel high-level rewards in hierarchical RL and combines this with HER so that relabeled rewards are (i) stationary, because they are computed solely by the LLM-defined reward function, and (ii) denser, via subgoal substitution from hindsight goals. The work explicitly targets non-stationarity in HRL with a “language-to-reward” pipeline and reports improved performance over baselines without language guidance.

**Strengths:**

- The problem (high-level non-stationarity) is clearly framed, with a straightforward idea: use LLM-generated reward functions to relabel the high-level buffer.

- Experiments/ablations are extensive (success curves, HRL+HER comparison, and a non-stationarity metric), with 5-seed reporting.

**Weaknesses:**

- Real-robot section / appendix cross-reference. The paper asks “Can LGR2 policies transfer to real-world robots?” and refers to “supplementary Section 9,” but the appendix appears to run A.1–A.7 with no Section 9/A.9 present after careful checking. The real-world description in the main text is also brief, which limits verification. Overall, the presentation should be further checked / improved.
- Non-stationarity metric detail. The metric is important (avg. distance between higher-level subgoals and states achieved by the lower level), but the “why/how” of the distance is not specified beyond captions; more detail would help interpretability.
- What exactly is generated. The motion-descriptor → reward-coder pipeline is described, but concrete final reward implementations are hard to find; prompts and “sample codes” remain toy-like (e.g., print statements). Including real reward modules would strengthen reproducibility.

**Questions:**

- In Figure 2, the proposed method achieves positive success rates while most baselines barely progress. Beyond the stated motivation (non-stationarity), could the authors show concrete cases of the non-stationary phenomenon and how L2R relabeling changes the outcome (e.g., brief case studies/visualizations)? This would clarify why reward relabeling yields such a large gap when non-stationarity may not always dominate.

---

### Official Review · Reviewer_gH8s · 2025-11-07

**Soundness:** 2
**Presentation:** 3
**Contribution:** 2
**Rating:** 4
**Confidence:** 5

**Summary:**

This paper presents LGR2, a hierarchical reinforcement learning framework designed to address the non-stationarity problem in off-policy HRL. The core contribution is the use of a Large Language Model to generate a stationary, language-guided reward function for the high-level policy. The method is combined with Hindsight Experience Replay to address sparse reward signals.

**Strengths:**

The paper tackles the problem of non-stationarity in off-policy HRL, which destabilizes training and impedes convergence. The proposed solution is clear in its application: using an LLM to generate a static reward function $r_{\phi}$ that depends only on state and goals, not the low-level policy's behavior. This decoupling approach is a sound strategy for mitigating reward drift. The experimental validation covers four simulated robotic environments and real-world transfer, and the inclusion of key ablations helps isolate the different components of the contribution.

**Weaknesses:**

Despite its strengths, the paper has several weaknesses that limit its impact:

1.  **Limited Novelty:** The primary concern is originality. The LGR2 framework appears to be a straightforward combination of two existing, major frameworks: standard HRL and the Language-to-Reward (L2R) pipeline proposed by Yu et al. (2023). The paper’s core idea is to apply the L2R's reward generation mechanism to the high-level policy in an HRL setup. While this combination is logical, the paper would be stronger if it more clearly articulated the novel technical advancements beyond this direct integration.

2.  **Unconvincing Experimental Baselines:** The experimental results, while impressive for LGR2, raise concerns due-to the performance of the baselines. In three of the four tasks ('Pick and place', 'Bin', and 'Kitchen'), virtually all baseline methods (including L2R, SAGA, HAC, and HIER) show near-zero success rates throughout training. This outcome is highly suspect. These tasks, while sparse, are standard in robotics, and it is unlikely that all modern HRL and language-conditioned baselines would fail completely. This suggests a potential issue with the implementation or hyperparameter tuning of the baselines, which makes the reported performance gap of LGR2 less convincing.

3.  **Weak Justification for Task Complexity:** The paper motivates the use of HRL by citing "long-horizon" and "complex" tasks. However, the chosen tasks, particularly 'Pick and place' and 'Bin', are not self-evidently "long-horizon" problems that *require* hierarchical decomposition and subgoals. The necessity of a hierarchical approach over a well-tuned flat policy (e.g., L2R + HER) is not sufficiently established for these environments.

4.  **Muddled Contribution of the LLM Reward:** The paper's motivation is split. It claims the LLM-guided reward $r_{\phi}$ solves non-stationarity. However, the ablation study in Figure 5 shows that LGR2 without HER (LGR2-No-HER) completely fails. This strongly implies that the LLM-generated reward $r_{\phi}$ is just as sparse as the original environment reward. If the LLM's "semantic richness" does not produce a denser or more shaped reward signal, its only function is to be *stationary*. This "stationary but sparse" signal seems to underutilize the LLM's capabilities and could potentially be replaced by a simple, hand-crafted, stationary reward function (e.g., goal-distance) combined with HRL+HER (the HIER-HER baseline).

**Questions:**

1.  **Baseline Performance:** Could the authors elaborate on the hyperparameter tuning process for the baselines (L2R, HAC, HIER, etc.)? Why do all of them achieve near-zero success on the manipulation tasks as shown in Figure 2? A more robust comparison against stronger, more recent language-conditioned or HRL baselines that show non-zero performance would be necessary to truly validate LGR2's superiority.

2.  **Task Complexity:** Can the authors provide metrics (e.g., average episode length, complexity of the required action sequence) to quantitatively justify the "long-horizon" nature of the 'Pick and place' and 'Bin' tasks? Why is a hierarchical approach fundamentally necessary for these tasks compared to a flat policy?

3. **Notational Clarity:** There is a confusing use of notation for the high-level reward, specifically between Section 4.1 and Section 4.2.
    * In Section 3, $r_t^H$ is defined as the *cumulative environment reward*.
    * In Section 4.1, the "Reward Coder Module" paragraph defines the language-guided reward function as $r^{H}=r_{\phi}(s,g^{*},g)$.
    * However, in Section 4.2, Equation (2) introduces a relabeled reward $r_{t}^{\varphi}$ , which is then defined as $r_{\varphi}(s_{t},g^{*},g_{t})$.
    * Are $r^H$ (from 4.1) and $r_t^{\varphi}$ (from 4.2) the same reward? The use of different base symbols ($H$ vs $\varphi$) and different subscripts ($\phi$ vs $\varphi$) is confusing. Please clarify if $r_{\phi}$ and $r_{\varphi}$ represent the same function, and if $r^H$ (from 4.1) is the same as $r_t^{\varphi}$ (from 4.2).

4.  **Sparsity of Language-Guided Reward:** Given that LGR2-No-HER fails, this implies the LLM-generated reward $r_{\phi}$ is sparse.
    * Does this mean the *only* benefit of the complex LLM-generation pipeline is to provide a *stationary* signal, rather than a *denser* or more *semantically meaningful* one?
    * How does LGR2's performance compare against the HIER-HER baseline (Figure 3) if HIER-HER were augmented with a simple, hand-crafted, *stationary* reward function (e.g., $r^H = -||s_{t+k} - g_t||_2$)? This would help isolate the true value of the *language-guidance* versus just *stationarity*.

---

### Official Review · Reviewer_y22P · 2025-11-12

**Soundness:** 3
**Presentation:** 2
**Contribution:** 2
**Rating:** 4
**Confidence:** 4

**Summary:**

The paper presents LGR2, a framework that uses language models (LLMs) to guide reward relabeling in hierarchical reinforcement learning (HRL). The approach introduces a language-guided reward function that leverages semantic information extracted from task descriptions, demonstrations, or subgoal prompts. Instead of relying on manually designed reward functions or extensive hindsight relabeling, LGR2 uses LLMs to generate context-aware goal descriptions and adjust reward signals accordingly.

The method is evaluated on hierarchical manipulation and navigation benchmarks, including tasks from BridgeData, MiniGrid, and CALVIN, showing gains in both sample efficiency and generalization over other HRL methods (HAC, HIER, RAPS) and recent LLM-assisted baselines (L2R). Results indicate up to +25% success rate improvement and faster convergence under sparse reward conditions.

**Strengths:**

- **Novel mechanism for reward shaping:** The paper identifies a key bottleneck in HRL as reward relabeling due to non-stationarity and provides a new solution that integrates language-based semantics into the process.
- **Improved sample efficiency:** Empirical results show that language-informed reward signals lead to faster policy convergence and more stable subgoal learning.
- **Strong empirical evidence:** Evaluations across three diverse domains with ablation studies isolating the effect of language guidance, subgoal granularity, and prompt quality.
- **Clear problem formulation:** The paper defines the reward relabeling mechanism precisely, aligning it with standard HRL decomposition (meta-controller and sub-policy).
- **Potential generality:** LGR2 could be extended to other settings where reward specification is ambiguous or underspecified.

**Weaknesses:**

- **Limited novelty relative to prior LLM-guided RL works:** Similar translator  to **L2R**, which also integrate language into reward or goal inference. The paper could articulate more clearly how LGR2 surpasses these beyond implementation differences.  L2R and Text2Reward are not mentioned in the related work.
- **Dependence on language accuracy:** Relabeling quality is tied to the correctness and consistency of LLM-generated feedback; noisy descriptions can distort reward signals.
- **Sparse theoretical grounding:** While empirical results are strong, the paper lacks an analytical discussion of why language guidance improves reward attribution or hierarchical credit assignment.
- **Limited real-world validation:** While the abstract mentioned real robot, all experiments seem to be in simulation; there is no physical-robot demonstration of robustness to perception noise.
- **Clarity of comparisons:** Some baselines (e.g. L2R) seem to be reimplemented by the authors; it would help to specify whether the same LLM backbones and prompts were used, why they behave worse in simpler environements like Maze.

**Questions:**

1. How does LGR2 differ fundamentally from **L2R**, which also use language cues for reward estimation?  Seems like the main advantage is that high-level policy is getting rewards for choosing 'right' low-level actions independent of their success. This is like stacking two L2R together.
2. How robust is reward relabeling when the LLM produces ambiguous or semantically incorrect feedback?
3. What is the computational overhead introduced by LLM querying during training?
4. Could the framework generalize to multi-agent or dynamic task hierarchies?
5. Why do you call reward relabelling? When is it being relabelled?

---

### Note · Authors · 2025-11-20

**Comment:**

Thank you very much for all reviewers' careful review and valuable comments on our paper. We think that our current method requires more comprehensive ablation studies to verify the effectiveness and necessity of each module. To ensure the rigor and completeness of our research work, we have decided to withdraw this submission. We will continue to improve our work and look forward to contributing to the community in the future.

**Withdrawal Confirmation:**

I have read and agree with the venue's withdrawal policy on behalf of myself and my co-authors.